# Induction of Inflammation Disrupts the Negative Interplay between STING and S1P Axis That Is Observed during Physiological Conditions in the Lung

**DOI:** 10.3390/ijms24098303

**Published:** 2023-05-05

**Authors:** Michela Terlizzi, Chiara Colarusso, Anna Falanga, Pasquale Somma, Ilaria De Rosa, Luigi Panico, Aldo Pinto, Piera Maiolino, Rosalinda Sorrentino

**Affiliations:** 1Department of Pharmacy (DIFARMA), University of Salerno, 84084 Salerno, Italy; ccolarusso@unisa.it (C.C.); afalanga@unisa.it (A.F.); aldopinto952@gmail.com (A.P.); 2Anatomy and Pathology Unit, Ospedale dei Colli, AORN, “Monaldi”, 84131 Naples, Italy; somma.pasquale@tiscali.it (P.S.); ilaria.derosa@ospedalideicolli.it (I.D.R.); lupanico59@gmail.com (L.P.); 3“Fondazione Pascale”, National Institute of Tumor, 80131 Naples, Italy; pieramaiolino15@gmail.com

**Keywords:** STING, sphingosine-1-phosphate (S1P), IL-6, inflammation, lung cancer

## Abstract

The stimulator of interferon genes (STING) is a master regulator of innate immunity, involved in several inflammatory diseases. Our previous data showed that sphingosine-1-phosphate (S1P) is released during inflammatory conditions in the lung. The aim of this study was to understand the interplay between S1P and STING during both physiological and pathological conditions. The mRNA levels of ceramidase (ASAH1), S1P precursor enzyme, and STING were inversely correlated in healthy lung tissues, but positively correlated in tumor tissues. The activation of STING induced higher expression of ASAH1 and was accompanied by IFN-β and IL-6 release. ASAH1 and sphingosine kinases (SPHK I/II) blockade significantly reduced IL-6, but not IFNβ, after STING activation. In support of this, taking advantage of a mouse model, we found that inflamed lungs had higher levels of inactive ASAH1 when STING was inhibited. This confirmed the human data, where higher levels of STING promoted the activation of ASAH1. Lung cancer patients positive to STING and ASAH1 mRNA levels had a dismal prognosis in that the overall survival was reduced compared to STING/ASAH1 negative patients. These data highlight that during physiological conditions, STING and the S1P axis do not interfere, whereas in lung cancer patients their interplay is associated to poor prognosis.

## 1. Introduction

The cyclic GMP-AMP synthase (cGAS) is the pivotal cytosolic DNA-sensor that catalyzes the synthesis of cyclic-guanosine monophosphate (GMP)-adenosine monophosphate (AMP)-nucleotide (cGAMP), which binds and activates the adaptor protein stimulator of interferon genes (STING). STING, also known as TMEM173, is an immune adaptor protein that governs signal crosstalk implicated in many physiological and pathological processes [1]. Although it has been established that STING traffics from the endoplasmic reticulum (ER) to the Golgi apparatus upon DNA derivatives recognition, emerging evidence reveals that STING can be transported to different organelles. This dictates its immune-dependent (e.g., the production of type I IFN and pro-inflammatory cytokines) and/or immune-independent (e.g., the activation of autophagy, ER stress, cell death, and lipid metabolism) physiological functions [2]. Upon activation and translocation, STING binds and activates TANK-binding kinase 1 (TBK1), ensuing in IFN regulatory factor 3 (IRF3) recruitment and phosphorylation. Phospho-IRF3 moves to the nucleus and promotes the transcription and synthesis of type I IFN [3]. However, STING also triggers NF-κB which in turn can induce the synthesis/release of pro-inflammatory cytokines, such as IL-6 [3].

In addition to its physiological functions, emerging evidence highlights the role for STING in human diseases where its variants as well as dysregulated signaling are involved in inflammation-driven pathological conditions [3,4,5]. An over-activation of STING in macrophages is associated to obesity-induced lung inflammation [6] ensuing in lung dysfunction. In support of this, a gain-of-function mutation of *STING1* is responsible for vasculopathy in infancy (SAVI), a type I interferonopathy characterized by severe inflammatory lung disease [7].

Therefore, identifying the mechanisms that can favor an alteration of the physiological role of STING, and which therefore can be associated with the development of chronic inflammatory diseases, could represent a turning point in research aimed at identifying molecular/therapeutic targets.

Sphingosine-1-phosphate (S1P) is a multifunctional bioactive lipid, involved in numerous physiological processes, such as chemotaxis, migration, growth, and proliferation [8]. It participates in organ homeostasis as in the case of the heart, lung, brain, and blood vessels [8,9]. S1P is a metabolite of ceramide, synthetized by two major mechanisms, a de novo synthesis and the sphingomyelinase-dependent pathway. Ceramide is converted into sphingosine by ceramidase (ASAH1), and in turn it is phosphorylated into S1P by two sphingosine-kinases (SPHKs), SPHK I and SPHK II [10]. S1P overproduction has been associated with the development of chronic inflammatory respiratory disorders [11,12,13,14,15]. Since it is known that STING is involved in lipid metabolism [16], in the regulation of the immune response of various pulmonary inflammatory dysfunctions [17], and in mediating the metabolic alterations associated to lung inflammation [6], the aim of this study was to investigate a potential crosstalk between STING and S1P signaling.

We found that STING-derived IL-6, but not IFN-β, was dependent on ceramidase as well as on SPHK I and II activation in lung epithelial cells. This mechanism, however, was only observed during pathological, but not physiological, conditions. Indeed, during physiological homeostasis, STING and S1P were not correlated.

## 2. Results

### 2.1. The Release of IL-6 after STING Activation Is S1P-Dependent in Lung Epithelial Cells

It is known that S1P is involved in inflammatory processes in the lung [15,18]. Instead, STING agonists seem to have promising potential anti-tumor activity [19]. The goal of this study was to understand whether the S1P-dependent pathway and STING were correlated. The stimulation of STING by means of cGAMP promoted the over-expression of the active form of ASAH1 (ceramidase, 40 kDa), the enzyme involved in S1P synthesis (Figure 1A), implying an interplay between the two pathways in lung tumor epithelial cells.

Because STING activation leads to the release of IRF3- and NF-κB-dependent cytokines [3], we measured the levels of IL-6 and IFN-β. The stimulation of lung tumor epithelial cells with cGAMP significantly increased both IFN-β (Figure 1B) and IL-6 (Figure 1C). To verify whether this effect was solely dependent on STING, STING was inhibited by H151. As expected, the blockade of STING significantly reduced the levels of IFN-β (Figure 1B) and IL-6 (Figure 1C).

In our previous studies, the stimulation of lung tumor epithelial cells with S1P induced the release of pro-inflammatory cytokines either directly or via TLR9 activation [15]. Therefore, to understand the interplay between the STING pathway and ASAH1 activation, mainly involved in sphingosine synthesis, we used a ceramidase inhibitor, D-NMAPPD. Interestingly, the inhibition of ceramidase did not alter IFN-β levels after STING activation (Figure 1B). In sharp contrast, the release of IL-6 was significantly reduced (Figure 1C), further supporting the hypothesis that the STING pathway and the sphingosine axis were interconnected.

Because the synthesis of S1P is strictly dependent on SPHK I/II activity, we inhibited either SPHK I or SPHK II by means of PF543 or Opaganib, respectively. The inhibition of SPHK I or SPHK II did not alter IFN-β levels after cGAMP stimulation (Figure 2A), but it significantly reduced the levels of IL-6 (Figure 2B).

These results demonstrate that the S1P axis is involved in STING-induced IL-6, but not in IFN-β, release.

### 2.2. S1P Axis and STING Pathway Are Not Connected in the Lung during Physiological Conditions

Previous data are related to epithelial tumor cells. In order to understand the physiological relationship between the S1P and STING axes, and because of the limitation of healthy lung tissue samples, we took advantage of a public RNAseq database (TCGA_LUAD_2016), separating lung normal tissues from tumor tissues, considering the normal tissues as during a physiological status.

The mRNA levels of both ceramidase (ASAH1, Figure 3A,B) and STING (Figure 3A,B) were present in normal lung tissues as reported by the public database (TCGA_LUAD_2016). It has to be noted that the expression of ceramidase was higher than STING (Figure 3A,B). In addition, in healthy, non-tumor samples, STING and ASAH1 were not correlated to each other (Figure 3C,D). It is noteworthy that although STING mRNA was lower than ASAH1 in terms of read counts, its expression was inversely correlated to the expression of ASAH1, implying that an increase in the activity of one could correspond to a decrease in the activity of the other, and vice versa (Figure 3C,D).

To understand the connection between STING and the S1P axis during physiological conditions, we stratified patients according to the median of read counts (expression) of ASAH1 (median = 5.849) and STING (median = 3.210). In particular, patients were stratified as follows: ASAH1+ (*n* = 27), ASAH1− (*n* = 25), STING+ (*n* = 26), and STING− (*n* = 26). A gene set enriched analysis (GSEA) was performed following the hallmark database.

The comparison of ASAH1+ vs. ASAH1− showed that ASAH1+ patients were characterized by higher metabolic pathways, such as oxidative phosphorylation, fatty acid metabolism, adipogenesis, cholesterol homeostasis, bile acid metabolism, heme metabolism, and glycolysis, and cell survival/proliferation pathways, such a MYC targets v1, MTORC1 signaling, and E2F targets (Figure 4A). Instead, STING+ patients had enriched pathways typical of immune system activation, such as IL6-JAK-STAT3 signaling, inflammatory responses, TNFA signaling via NF-κB, IL2-STAT5 signaling, and complement, and enriched cell growth-related pathways, such as epithelial mesenchymal transition, MYC target v2, P53 pathway, and KRAS signaling UP, compared to STING− patients (Figure 4B). The plots of the 15 most enriched gene sets are shown in Appendix A (for ASAH1+ vs. ASAH1−) and Appendix A (for STING+ vs. STING−).

These data reveal that the S1P axis and STING pathway are inversely linked under physiological conditions in the lung and that the predominance of the sphingosine axis during physiological conditions is mainly deputed to metabolic programs, whereas STING is more related to the physiological activation of the immune system. In support of this, in our previous studies, normal fibroblasts, used as an example of healthy non-cancerous cells, were not able to release inflammatory cytokines under S1P/axis stimulation, differently than cancer cells which were responsive to S1P stimulation in terms of pro-inflammatory cytokine release [15].

### 2.3. S1P Axis and STING Pathway Are Interconnected during Pathological Conditions in the Lung

Similarly to what was performed on normal lung tissues, we analyzed the public TCGA_LUAD_2016 database to evaluate both STING and ASAH1 mRNA levels in tumor tissues. Again, STING mRNA was lower than ASAH1 levels (Figure 5A,B) during pathological conditions, too. However, in sharp contrast to the physiological status, tumor lung tissues showed that higher levels of STING were associated to higher levels of ASAH1 (Figure 5C,D). Indeed, we found a positive correlation between STING and ASAH1 expression (*p* < 0.0001, r = 0.2289, R2 = 0.05), implying that during pathological conditions the two are upregulated and most likely may participate in common pathways, as observed in in vitro studies (Figure 2).

To understand their crosstalk in lung cancer, a GSEA analysis was performed according to the hallmark database, stratifying patients according to the median expression of ASAH1 (median = 4.643) and STING (median = 2.767) in lung tumor tissues. Patients were stratified as follows: ASAH1+ (*n* = 152), ASAH1− (*n* = 147), STING+ (*n* = 141), and STING− (*n* = 158). GSEA analysis showed that tissues of ASAH1+ patients were mainly characterized by metabolic pathways (fatty acid metabolism, bile acid metabolism, heme metabolism, adipogenesis, cholesterol homeostasis) compared to ASAH1− patients (Figure 6A), as observed in non-tumor tissues (Figure 4A). The same was observed for STING+ patients, who were mostly characterized by immune system-related (inflammatory response, TNFA signaling via NF-κB, IL6-JAK-STAT3 signaling, interferon gamma and alpha response, IL2-STAT5 signaling, complement, TGF-beta signaling) and cell growth-related pathways (epithelial mesenchymal transition, P53 pathway, KRAS signaling UP) compared to STING− patients (Figure 6B). To note, a difference between STING+ healthy versus tumor tissues was observed (Figure 3 vs. Figure 4B). Plots of the 15 most enriched gene sets are shown in Appendix A (for ASAH1+ vs. ASAH1−) and Appendix A (for STING+ vs. STING−).

Although there were similarities in the two pathways during physiological and pathological conditions, it has to be noted that the expression of STING when the tumor was present was positively correlated to the expression of ASAH1.

To understand at what stage STING and ASAH1 were correlated, we took advantage of a mouse model of carcinogen-induced lung cancer, in which chronic inflammation is at the basis of the malignant transformation [20]. N-Methyl N-Nitroso Urea (NMU), a carcinogen alkylating DNA agent, was instilled into C57Bl/6N mice intratracheally, while the STING inhibitor (H151) was i.p. administrated (please refer to the Section 4 for the experimental protocol). After euthanasia, the lungs were collected and homogenized, and the expression of ceramidase was evaluated. We found that in NMU+H151-treated mice, ASAH1 was over-expressed in its inactive form (55 kDa) compared to the mice exposed to the sole NMU (Figure 4A,B). No differences between the two groups in the ASAH1 active form expression were observed (Figure 4A,C). These data imply that STING could alter S1P generation. Thus, to evaluate a co-localization of both targets, we evaluated the expression of S1P and STING in non-cancerous, healthy, and cancerous human lung tissues. We found that S1P (red fluorescence) was bound to STING (green fluorescence) only in cancerous tissues but not in healthy non-tumor tissues (Figure 7D, merge of green and red fluorescence).

All together, these data suggest that although not during physiological conditions, in lung tumor tissues the STING and S1P axes may interplay.

### 2.4. The Co-Expression of STING and S1P Increased Lung Cancer Mortality

To understand whether the interplay between STING and S1P in lung cancer tissues was associated with patients’ overall survival, we considered the median values of ASAH1 (median = 4.643) and STING (median = 2.767) transcripts in lung tumor tissues as reported in the TCGA_LUAD_2016 database. Accordingly, lung adenocarcinoma patients were stratified as ASAH1+ (median > 4.643), ASAH1− (median < 4.643), STING+ (median > 2.767), and STING− (median < 2.767). We found that there were no differences in terms of overall survival between ASAH1+ and ASAH1− (Figure 8A) and between STING+ and STING− (Figure 8B) patients. These data prompted us to hypothesize that neither ASAH1 nor STING activity were relevant in tumor progression and survival, but when patients were stratified according to both target positivity or negativity, we found that lower expression of ASAH1 was associated to higher survival in STING+ patients (Figure 8C, ASAH1-STING+ 87 months vs. ASAH1+STING+ 59 months; *p* = 0.0475). In contrast, higher expression of ASAH1 and STING was associated to a significantly diminished overall survival rate (Figure 8C, blue line).

Therefore, these data further suggest that the S1P axis and STING-dependent pathways are both involved in lung adenocarcinoma progression and one depends on the other, although S1P axis activation is deleterious for STING positive patients. Nevertheless, it has to be noted that the limitation of this analysis stands in the evaluation of transcripts (TCGA-LUAD) rather than proteins and thus it is not possible to discriminate the active from the inactive form of ASAH1.

## 3. Discussion

STING-mediated innate immunity plays a significant role in shaping host defense against microbe invasion and tumor growth. However, aberrant activation of STING can also alter immune balance, thereby leading to pathological conditions and human diseases [21,22].

In this study, we found that in tumor cells the activation of STING by means of cGAMP resulted in an increased activation of ASAH1 responsible for S1P synthesis, as already reported [15,23]. The inhibition of STING reduced IL-6 release in in vitro studies, supported by in vivo studies. The lungs of mice undergoing NMU treatment had higher expression of the inactive form of ceramidase when STING was pharmacologically inhibited, implying that STING could interfere with S1P synthesis. Similarly, the inhibition of ceramidase as well as SPHK I/II in vitro abrogated cGAMP/STING-dependent IL-6, but not IFNα, release. These data prompted us to hypothesize that the S1P axis and STING-dependent pathways interplay in lung tumors. To prove this hypothesis, we used a public database of RNAseq and found that STING and S1P were co-expressed and were positively correlated to each other in lung tumor tissues compared to non-cancerous tissues (that we identified as representative of a physiological lung condition). ASAH1+STING+ patients had a lower survival rate than ASAH1-STING+ adenocarcinoma patients. However, it should be noted that only when the transcription of ASAH1 was high together with STING, then the survival rate was poorer. Thus, this could lead us to suppose that the activation of ASAH1 worsens lung cancer patients’ survival, mainly due to the activation of fatty acid metabolism, which we already demonstrated to be altered in lung cancer [24], as well as proto-oncogene pathways (Figure 6). Indeed, from bioinformatic analysis, it emerges that the predominance of the sphingosine axis during both physiological and pathological conditions is mainly deputed to the lipidomic rearrangement, which is well known to affect the immune system [24]. Thus, we could further hypothesize that S1P is a lipid mediator capable of altering the physiological status and that together with STING activation can lead towards a chronic inflammatory pattern at the basis of tumor establishment/progression. However, this hypothesis needs further elucidation.

We previously demonstrated that S1P is involved in the exacerbation of the TLR9-induced inflammatory pathway associated to lung cancer [15,25], as well as in cell proliferation through a SPHK II/S1PR3 nuclear-dependent intracellular mechanism [26]. Here, cGAMP-induced STING activation induced an over-expression/over-activation of ceramidase (ASAH1) and was associated to an amplification of IL-6 release, a cytokine which correlates with JAK and STAT3/NF-κB-dependent patterns, which were found to be enriched by using machine learning data (Figure 6B) in STING+ patients. However, a recent study demonstrated that SPHK II-generated S1P in CD11b+ macrophages blocked STING, suppressing the inflammatory function of alveolar macrophages and resolving Pseudomonas aeruginosa-induced lung vascular injury [27]. The discrepancy with this study could be based on the pathological context (acute lung injury vs. lung cancer), as well as on the cellular nature (immune circulating cells vs. epithelial cells). Moreover, Joshi and colleagues demonstrated the involvement of S1P only in blocking the release of IFN-β and not of IL-6. In fact, from our data it emerges that S1P is not involved in the release of STING/IRF3-mediated IFN-β, but that S1P intervenes in the other downstream STING pathway, which was NF-κB-dependent. In support of our data, a recent study showed a direct role for STING signaling in IL-6 production in response to the genotoxic treatment of cancer cells, independent of an IFN signature, biasing STING activity towards a pro-tumorigenic profile [28]. In line with this, our data provide a new mechanism that involves S1P as a driver of STING function in the lung.

However, the mechanism by which STING is able to induce the activation of ceramidase, and therefore an over-production of S1P, remains to be elucidated. It is certain that when STING is inhibited in a context of chronic lung inflammation, as in the case of our experimental model of carcinogen-induced lung cancer, ceramidase is not activated. This could likely be explained by the fact that STING induces S1P metabolism. This question could find an answer because of the co-localization of STING with acetyl-CoA carboxylase (ACC) and fatty acid synthase (FASN), two important enzymes of de novo lipid biosynthesis, suggesting that they might be components of a multi-protein complex involved in fatty acid generation [16], as highlighted by GSEA analysis (Figure 4 and Figure 6). In fact, we also found that in cancerous lung tissues, STING and ASAH1 transcripts were linked by a positive correlation, implying that an increase in the activity of one corresponds to an increase in the activity of the other, and vice versa, supporting the idea that these two pathways are strictly related in tumor tissues. In line with this, we demonstrated that the protein STING binds S1P in lung cancer tissues, but not in non-cancerous tissues, suggesting a possible crosstalk prevailing in pathological conditions, like lung cancer, rather than in physiological conditions. In support of this, another study demonstrated that S1P interacts with STING in bone marrow-derived macrophages (BMDMs) [27]. These researchers proved that S1P can interact with STING in three different modes, depending on whether STING is in a closed or open conformation. S1P can occupy the binding pocket of cGAMP when STING is in the closed conformation. In contrast, S1P has two possible binding modes within the open conformation of STING: it can occupy the c-di-GMP binding site in a competitive mode or it can bind to an allosteric site that is at the interface between the two monomers of STING [27]. Further studies are therefore needed to understand when STING prevails in one conformation rather than the other, and how STING functions change in relation to the different binding modes of S1P. Despite these gaps, however, we found that the co-expression/co-activation of both STING and ASAH1 in lung cancerous tissues predicted a worsened survival rate compared to patients who were only STING+ (ASAH1+ STING+ 59 months vs. ASAH1− STING+ 87 months; *p* = 0.0475), confirming that S1P can exacerbate patients’ prognosis.

In conclusion, our study identifies S1P as another mediator that together with STING can facilitate lung tumor progression and poor survival. Therefore, because STING represents nowadays an emerging therapeutic target, the use of agonists in anti-cancer therapy must be carefully considered, especially in relation to STING’s ability to enhance S1P signaling.

## 4. Materials and Methods

### 4.1. Cell Cultures

Human lung epithelial A549 cells line (ATCC^®^ CCL-185™) were cultured in Dulbecco’s Modified Eagle Medium (DMEM) (Cambrex Biosciences, Microtech, Naples, Italy) supplemented with 10% fetal bovine serum (FBS), 100 units/mL penicillin, 100 units/mL streptomycin, and 2 mM L-glutamine (Cambrex Biosciences, Microtech, Naples, Italy), in an atmosphere of 5% CO_2_ at 37 °C. Cells were seeded (3 × 10^3^/well) and treated with the following: cGAMP [2’-3’-cyclic guanosine monophosphate (GMP)—adenosine monophosphate (AMP)], the STING ligand (cGAMP 10 μg/mL; InvivoGen, Toulouse, France; #1441190-66-4), H151 [N-(4-Ethylphenyl)-N’-1H-indol-3-yl-urea], the STING inhibitor (H151 10 μg/mL; InvivoGen, Toulouse, France; #941987-60-6), D-NMAPPD, the ceramidase inhibitor (D-NMAPPD 5 µM; Sigma-Aldrich, Merck Life Science S.r.l., Milan, Italy; #SML2358), PF-543, a sphingosine-competitive inhibitor of sphingosine kinase I, SPHK I (PF543, 2 µM; Selleck Chemical, Houston, TX, USA; #57177), ABC294640 (Opaganib), a selective inhibitor of sphingosine kinase II, SPHK II (Opaganib, 60 µM; RedHill Biopharma, Tel-Aviv, Israel; #915385-81-8). The experimental time points were chosen upon preliminary time-dependent data in which we performed experiments at 1, 3, 5, 8, or 18 h. Specifically, the cells were treated for 3 h to evaluate the expression/activation of the enzymes involved in S1P metabolism, (i.e., ceramidase, ASAH1), through Western blotting assays as described below. Instead, after 8 h of treatment the release of STING activation-related cytokines was evaluated, as described below.

### 4.2. Cytokine Measurements

IFN-β (R&D System Inc., Minneapolis, MN, USA) and IL-6 (Diaclone SAS, Besançon, France) were measured in cell-free supernatants, obtained after 8 h of cell treatment, by means of a commercially available enzyme-linked immunosorbent assay kit (ELISAs). The absorbance wavelength was 450 nm. Cytokine levels were expressed as pg/mL.

### 4.3. Western Blotting Analysis

The expression of ceramidase (ASAH1, N-acylsphingosine amidohydrolase 1; active form: 40 kDa; precursor form: 55 kDa; Elabscience; Houston, TX, USA; #E-AB-10959) was evaluated in cGAMP 3-hour-treated lung epithelial cells; also, ceramidase expression was evaluated in lung homogenates of NMU- and NMU+H151-exposed mice (refers to mouse model below). Heat shock cognate protein 70 (HSC70; 70kDa; OriGene Technologies, Rockville, MD, USA; #TA332519) was used as loading control. Data were analyzed by means of ImageJ software 1.53a http://imagej.nih.gov/ij (accessed on 1 October 2022) (NIH, Bethesda, MD, USA).

### 4.4. Mouse Model

Female C57Bl/6N mice (6–8 weeks of age) (Charles River Laboratories, Lecco, Italy) were fed with a standard chow diet and maintained in specific pathogen-free conditions at the animal care facility of the Department of Pharmacy, University of Salerno. Mice were anesthetized with isoflurane and intratracheally (i.t.) instilled with N-Methyl N-Nitroso Urea (NMU), alkylating and mutagen agent, exhibiting its toxicity by transferring its methyl group to nucleobases in nucleic acids, which can lead to AT:GC transition mutations, for 16 consecutive weeks, starting with a high dose of 50 μg/mouse (in 10 μL of saline) at week 1, 8, and 12 followed by other administrations of 10 μg/mouse at week 2, 3, 9, 10, 13, 14. Mice were divided into the following groups: 1. NMU, i.t. instilled with NMU and 2. NMU+H151, i.t. instilled with NMU + intraperitoneally (i.p.) treated with H151, the STING inhibitor (10 μg/mice, administrated 2 times a week, after each NMU instillation). Lungs were isolated and the right lung lobes were digested with 1 U/mL collagenase (Sigma Aldrich, 3050 Spruce Street, St. Louis, MO, USA; Cat#C0130), and Western blotting assays were performed. This study was carried out in strict accordance with the recommendations in the Guide for Care and Use of Laboratory Animals of the Health National Institute. The experimental protocol was approved by the Ethical Committee for Animal Studies of the University of Salerno and Health Ministry with the approval number 13786/2014. All animal experiments were performed under protocols that followed the Italian (D.L. 26/2014) and European Community Council for Animal Care (2010/63/EU). Experiments were repeated twice. Each group was composed of 5 mice.

### 4.5. Correlation Analysis of Transcripts Expression

The Lung Cancer Explorer (LCE) online tool was applied (https://lce.biohpc.swmed.edu/lungcancer/, accessed on 1 October 2022) to evaluate the correlation between STING and ASAH1 in lung cancer patients. Specifically, the TCGA_LUAD_2016 RNAseq dataset, a lung database, consisting of 570 total samples of lung adenocarcinoma (52 normal samples and 518 tumor samples) was analyzed.

### 4.6. Survival Analysis

The RNAseq dataset of lung tissues (TCGA_LUAD_2016) was used to calculate the median expression of ASAH1 (median = 4.643) and STING (median = 2.767) in lung tissues of cancer patients. According to the median, the patients were stratified in the following groups: ASAH1+ (median > 4.643; *n* = 250), ASAH1− (median < 4.643; *n* = 246), STING+ (median > 2.767; *n* = 248), STING− (median < 2.767; *n* = 248), ASAH1+STING+ (*n* = 139), and ASAH1-STING+ (*n* = 109), and the survival curve was created.

### 4.7. Gene Set Enrichment Analysis (GSEA)

According to median expression of ASAH1 and STING transcripts in healthy (ASAH1 median = 5.849; STING median = 3.210) and cancerous (ASAH1 median = 4.643; STING median = 2.767) lung tissues obtained from the TCGA_LUAD_2016 RNAseq dataset, a GSEA analysis was performed using the hallmark database (h.all.v.7.4.symbols.gmt), with the following settings: number of permutation: 1000; collapse/remap to gene symbol: collapse; permutation type: gene set; chip platform: Human_NCBI_Gene_ID_MSignDB.v2022.1.Hs.chip; enrichment statistic: weighted; metric for ranking gene: Signal2Noise; gene list sorting mode: real; gene list ordering mode: descending; max size: 500; min size: 15.

### 4.8. Human Samples

Lung cancer patients were recruited at the “Monaldi-Azienda Ospedaliera (AORN)-Ospedale dei Colli” Hospital in Naples, Italy, according to the Review Board which approved the project and the patients’ informed consent. The experimental protocol was performed in accordance with the guidelines and regulations provided by the Ethical Committee of the hospital (protocol no. 1254/2014). The age of enrolled patients had a mean of 50 ± 10 years old. Lung tissues were collected during surgery; a portion of tissue coming from the tumor mass (here identified as ‘cancerous’) and one coming from a distal part of the lung of the same patient not affected by the tumor lesion (here identified as ‘non-cancerous’) were used to evaluate the expression of STING and S1P, as described below.

### 4.9. Immunofluorescence Analysis

Non-cancerous and cancerous lung tissues were analyzed for the presence of FITC-conjugated STING (TMEM173/STING Rabbit Polyclonal antibody, diluted 1:500; Proteintech, Rosemont, IL, USA; #19851-1-AP) and PE-conjugated S1P (Sphingosine-1-Phosphate Antibody, diluted 1:50; Echelon Biosciences Inc., Salt Lake City, UT, USA; #Z-P300). To stain cell nuclei, DAPI (4′,6-diamidino-2-phenylindole) was used. Images were observed by means of Carl Zeiss confocal microscopy (magnification: 40×).

### 4.10. Statistical Analysis

Data are reported as median and represented as scatter dot plots. Statistical differences were assessed with two-tailed Mann–Whitney U test or with one-way ANOVA followed by Tukey’s post hoc test, as specified in figure legends. For the correlation analysis, the simple linear regression was applied. *p* values less than 0.05 were considered significant. The statistical analysis was performed by using GraphPad prism 9.4.0 version (San Diego, CA, USA).

## Figures and Tables

**Figure 1 ijms-24-08303-f001:**
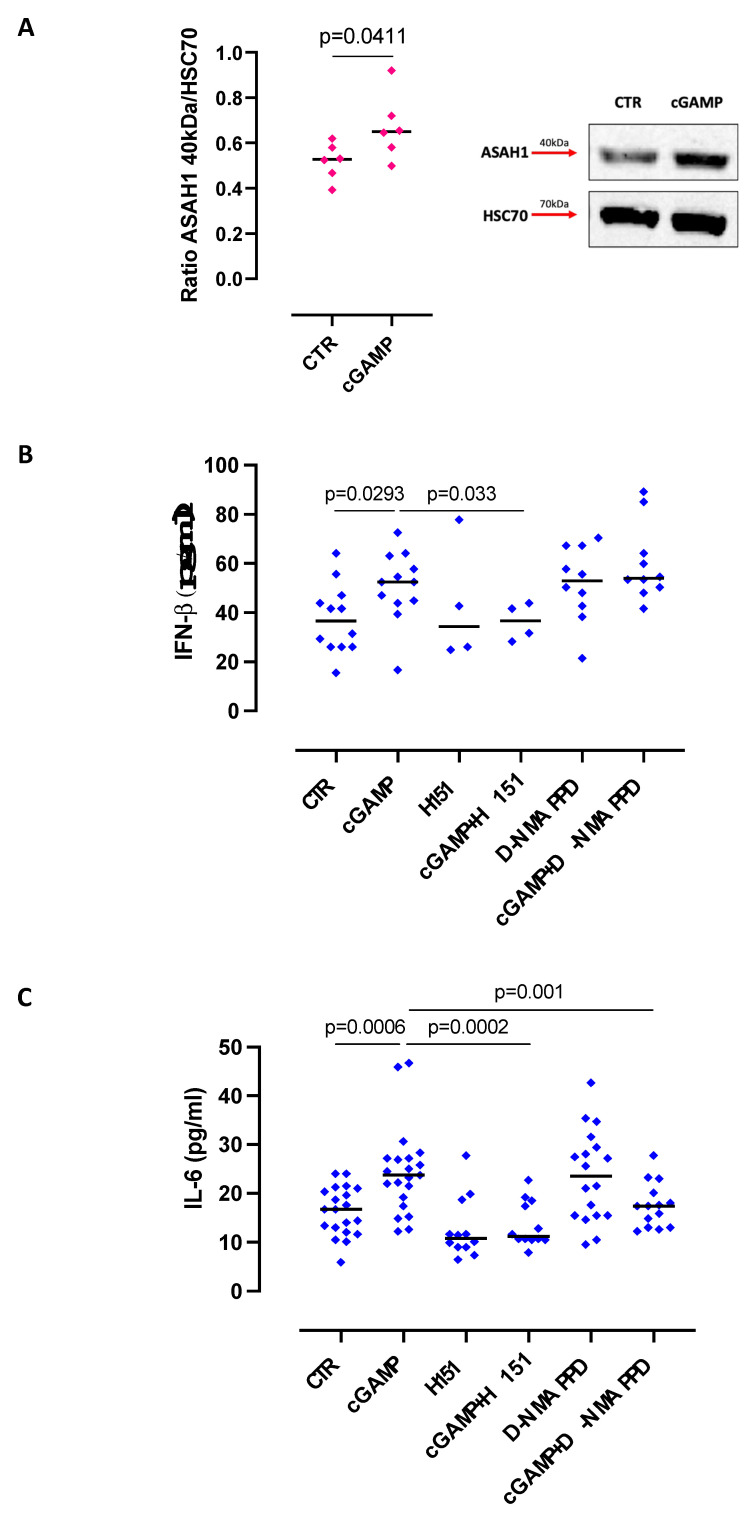
cGAMP-induced IL-6, but not IFN-β, release is ceramidase/S1P-dependent. Lung tumor epithelial cells were stimulated with cGAMP (10 μg/mL) for 3 h, and ceramidase (ASAH1) expression was evaluated by means of Western blotting. (**A**) cGAMP induced the over-expression of ASAH1 in its active form (40 kDa). HSC70 was used as a loading control. The quantitative analysis was performed by ImageJ software (NIH, USA). Lung tumor epithelial cells were stimulated with cGAMP (10 μg/mL) for 8 h, and the release of IFN-β (**B**) and IL-6 (**C**) was evaluated in the presence or not of H151, a STING inhibitor (10 μg/mL), or D-NMAPPD, an ASAH1 inhibitor (5 µM). Data are represented as scatter dot plots indicating the median (confidence interval = 95%). Statistical differences were assessed by means of two-tailed Mann–Whitney U test (**A**) and one-way ANOVA followed by Tukey’s post hoc test (**B**,**C**).

**Figure 2 ijms-24-08303-f002:**
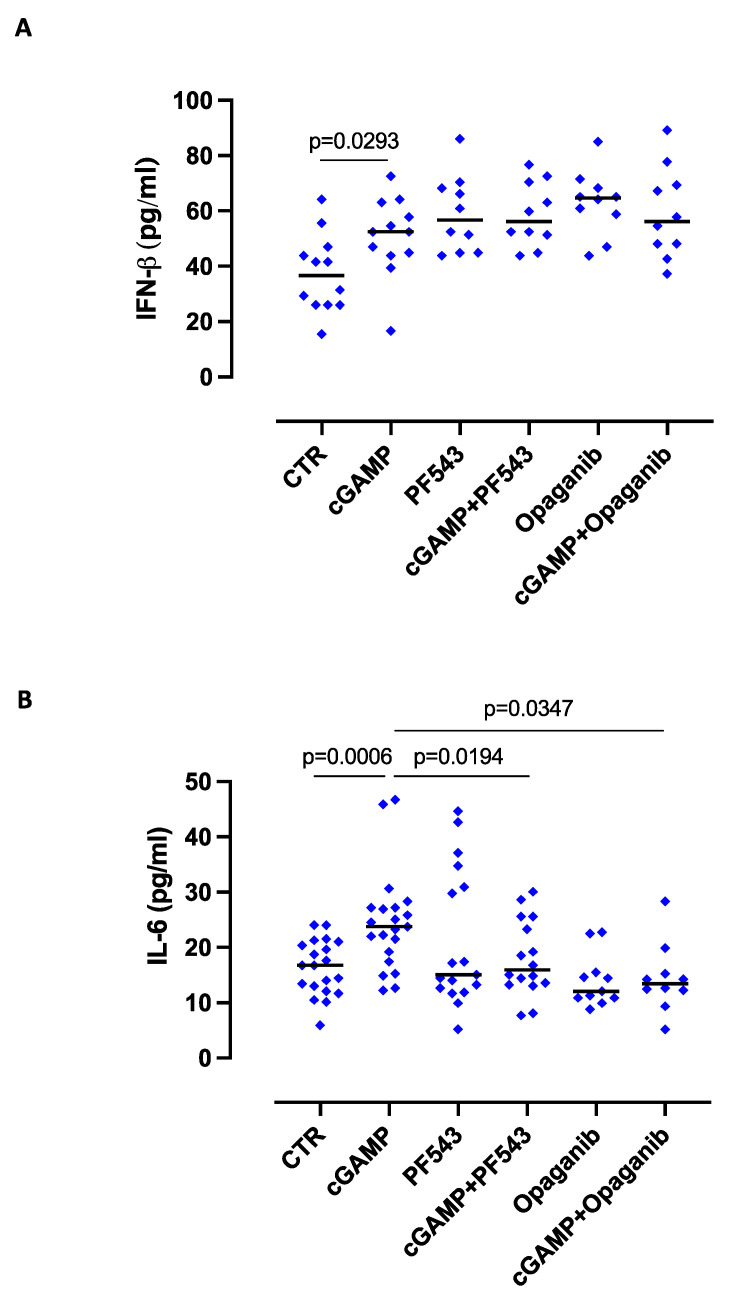
S1P is involved in the release of IL-6, but not of IFN-β, induced by cGAMP. Lung tumor epithelial cells were stimulated with cGAMP (10 μg/mL) for 8 h, and the release of IFN-β and IL-6 was evaluated in the presence or not of PF-543, a sphingosine-competitive inhibitor of sphingosine kinase I, SPHK I (2 µM), and ABC294640 (Opaganib), a selective inhibitor of sphingosine kinase II, SPHK II (60 µM). The inhibition of SPHK I or SPHK II did not alter the release of IFN-β (**A**), while it significantly reduced the release of IL-6 (**B**) after cGAMP addition. Data are represented as scatter dot plots indicating the median (confidence interval = 95%). Statistical differences were assessed by means of one-way ANOVA followed by Tukey’s post hoc test.

**Figure 3 ijms-24-08303-f003:**
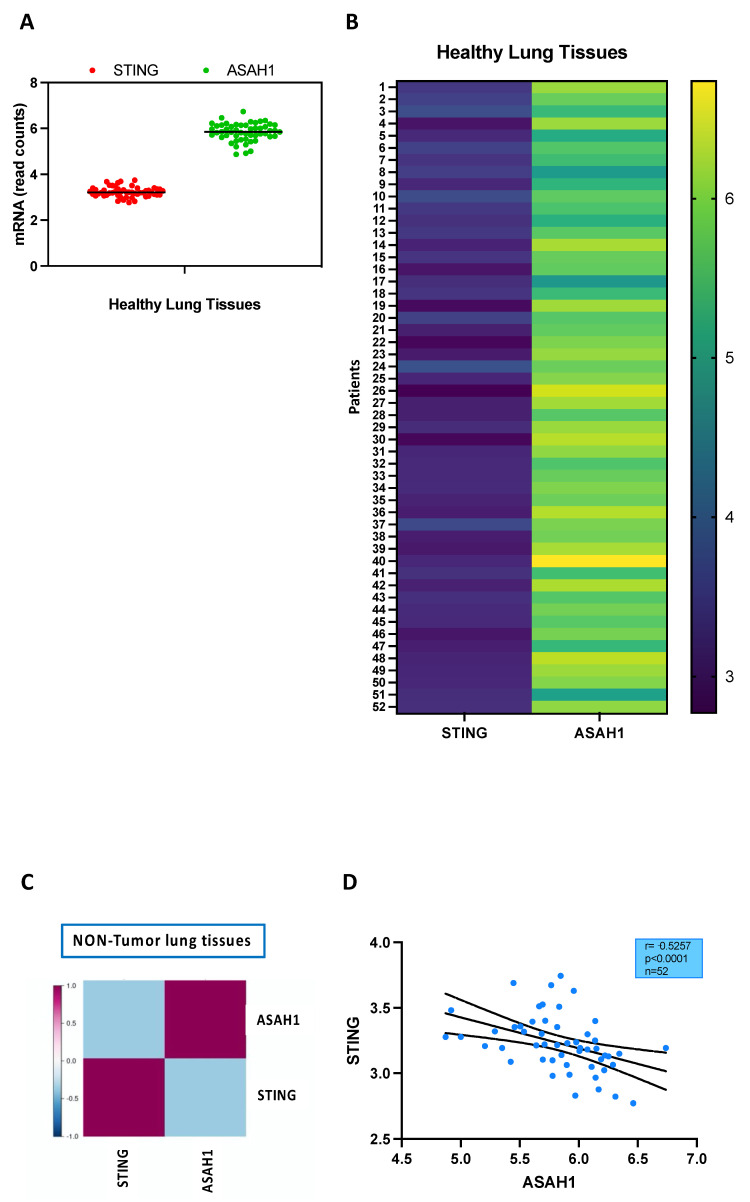
STING and ASAH1 are not correlated in healthy lung tissues. RNAseq data obtained by TCGA_LUAD_2016 database shows that the levels of ASAH1 mRNA (green dots) are higher than STING (red dots) in healthy lung tissues (*n* = 52) (**A**). (**B**) Heatmap representing the expression of STING and ASAH1 transcripts in healthy lung tissues (*n* = 52). The numbers of patients are shown on the y axis. A correlation analysis was performed through Lung Cancer Explorer (LCE) tool. Dendrogram showing the clustering of the genes (ASAH1 and STING) by similarity of expression profiles in NON-tumor lung tissues (**C**). (**D**) STING is negatively (Spearman r: −0.5257; 95% CI = −0.7026 to −0.2875; *p* < 0.0001) correlated to ASAH1 in healthy, NON-tumor lung tissues (*n* = 52).

**Figure 4 ijms-24-08303-f004:**
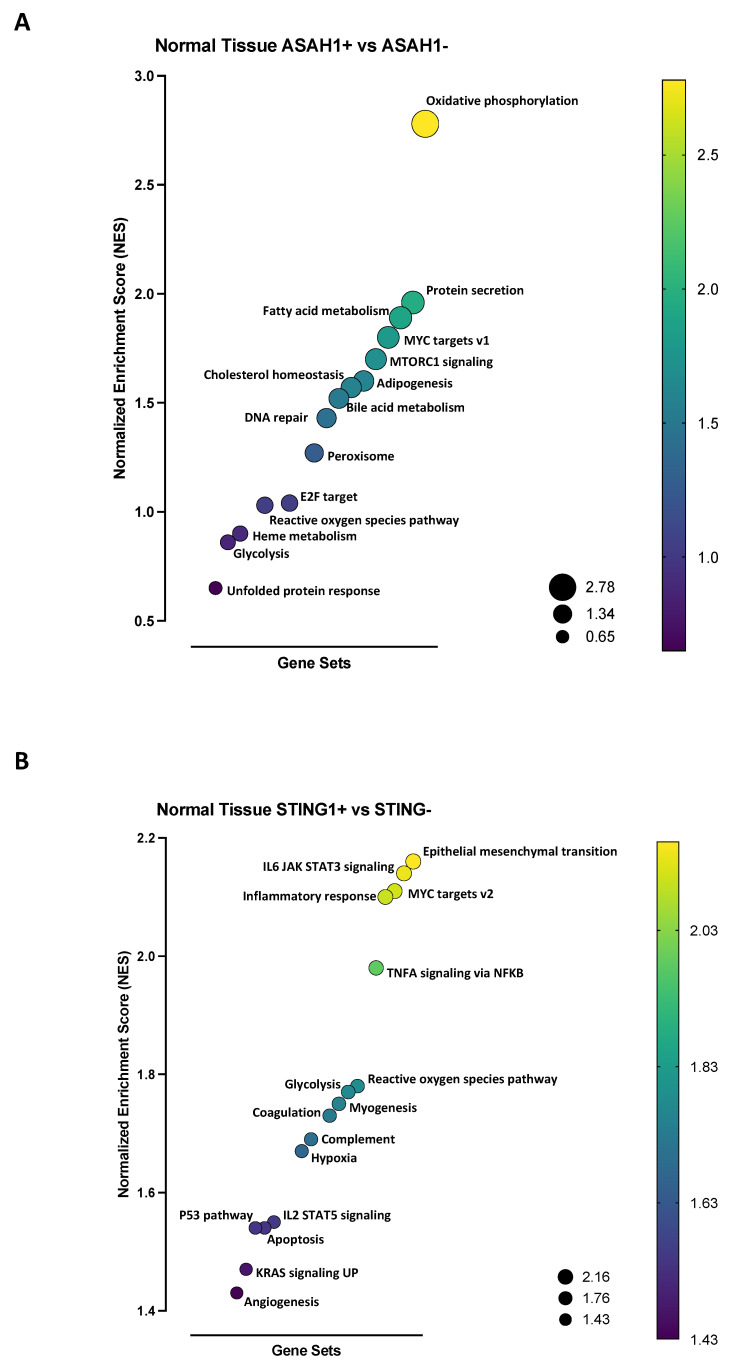
Gene set enriched analysis (GSEA) in healthy lung tissues. GSEA analysis was performed according to the hallmark database. Patients were stratified as ASAH1+ (*n* = 27), ASAH1− (*n* = 25), STING+ (*n* = 26), and STING− (*n* = 26) according to the median of read counts (expression) of ASAH1 (median = 5.849) and STING (median = 3.210). (**A**) Gene sets enriched in ASAH1+ compared to ASAH1− lung healthy tissues: oxidative phosphorylation (normalized enrichment score, NES = 2.78), protein secretion (NES = 1.96), fatty acid metabolism (NES = 1.89), MYC target v1 (NES = 1.8), MTORC1 signaling (NES = 1.7), adipogenesis (NES = 1.6), cholesterol homeostasis (NES = 1.57), bile acid metabolism (NES = 1.52), DNA repair (NES = 1.43), peroxisome (NES = 1.27), E2F targets (NES = 1.04), reactive oxygen species pathway (NES = 1.03), heme metabolism (NES = 0.9), glycolysis (NES = 0.86), unfolded protein response (NES = 0.65). (**B**) Gene sets enriched in STING+ compared to STING− lung healthy tissues: epithelial mesenchymal transition (NES = 2.16), IL6 JAK STAT3 signaling (NES = 2.14), MYC targets v2 (NES = 2.11), inflammatory response (NES = 2.1), TNFA signaling via NF-κB (NES = 1.98), reactive oxygen species pathway (NES = 1.78), glycolysis (NES = 1.77), myogenesis (NES = 1.75), coagulation (NES = 1.73), complement (NES = 1.69), hypoxia (NES = 1.67), IL2 STAT5 signaling (NES = 1.55), apoptosis (NES = 1.54), P53 pathway (NES = 1.54), KRAS signaling UP (NES = 1.47), angiogenesis (NES = 1.43). The bubble size is according to the NES.

**Figure 5 ijms-24-08303-f005:**
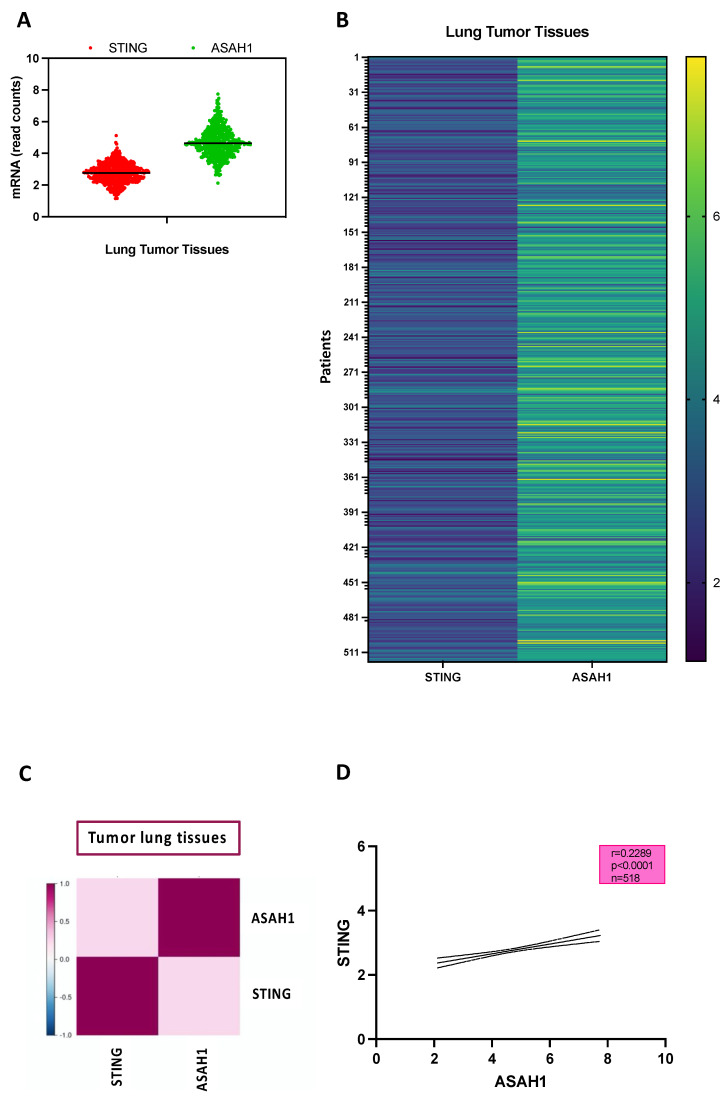
STING and ASAH1 are positively correlated in lung tumor tissues. RNAseq data obtained by TCGA_LUAD_2016 database show that the transcription of ASAH1 (green dots) is higher than STING (red dots) in lung tumor tissues (*n* = 518) (**A**). (**B**) Heatmap representing the transcripts expression of STING and ASAH1 in tumor tissues (*n* = 518). The numbers of patients are shown on the y axis. A correlation analysis was performed through Lung Cancer Explorer (LCE) tool. Dendrogram showing the clustering of the genes (ASAH1 and STING) by similarity of expression profiles in tumor lung tissues (**C**). (**D**) STING is positively (Spearman r: 0.2289; 95% CI = 0.1431 to 0.3112; *p* < 0.0001) correlated to ASAH1 in lung tumor tissues (*n* = 518).

**Figure 6 ijms-24-08303-f006:**
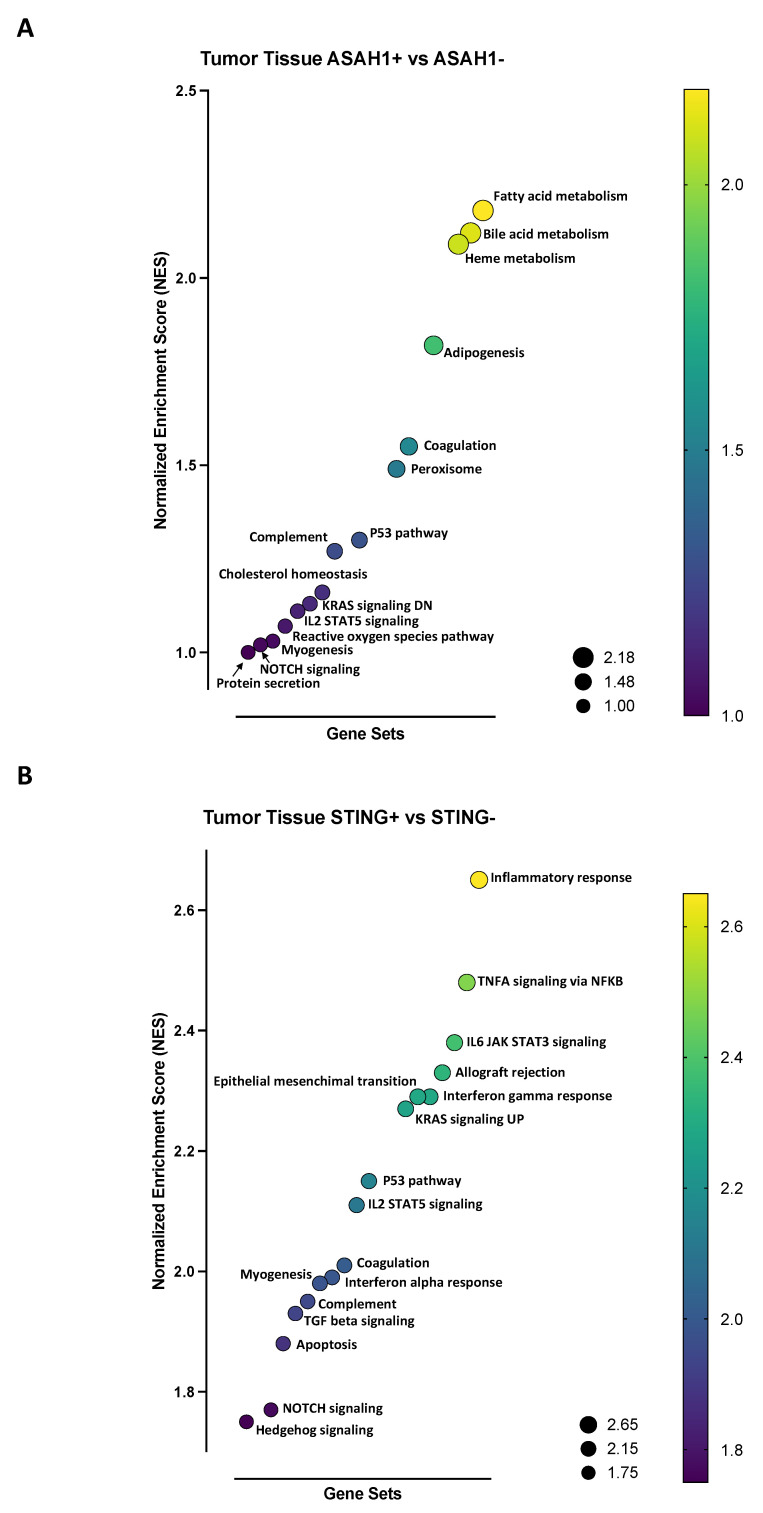
Gene set enriched analysis (GSEA) in tumor lung tissue. GSEA analysis was performed according to the *hallmark* database, stratifying patients according to the median expression of ASAH1 (median = 4.643) and STING (median = 2.767) in lung tumor tissues. Patients were stratified as ASAH1+ (*n* = 152), ASAH1− (*n* = 147), STING+ (*n* = 141), and STING− (*n* = 158). (**A**) Gene sets enriched in ASAH1+ compared to ASAH1− lung tumor tissues: fatty acid metabolism (normalized enrichment score, NES = 2.18), bile acid metabolism (NES = 2.12), heme metabolism (NES = 2.09), adipogenesis (NES = 1.82), coagulation (NES = 1.55), peroxisome (NES = 1.49), P53 pathway (NES = 1.3), complement (NES = 1.27), cholesterol homeostasis (NES = 1.16), KRAS signaling DN (NES = 1.13), IL2 STAT5 signaling (NES = 1.11), reactive oxygen species pathway (NES = 1.07), myogenesis (NES = 1.03), NOTCH signaling (NES = 1.02), protein secretion (NES = 1.00). (**B**) Gene sets enriched in STING+ compared to STING− lung tumor tissues: inflammatory response (NES = 2.65), TNFA signaling via NF-κB (NES = 2.48), IL6 JAK STAT3 signaling (NES = 2.38), allograft rejection (NES = 2.33), interferon gamma response (NES = 2.29), epithelial mesenchymal transition (NES = 2.29), KRAS signaling UP (NES = 2.27), P53 pathway (NES = 2.15), IL2 STAT5 signaling (NES = 2.11), coagulation (NES = 2.01), interferon alpha response (NES = 1.99), myogenesis (NES = 1.98), complement (NES = 1.95), TGF beta signaling (NES = 1.93), apoptosis (NES = 1.88), NOTCH signaling (NES = 1.77), hedgehog signaling (NES = 1.75). The bubble size is according to the NES.

**Figure 7 ijms-24-08303-f007:**
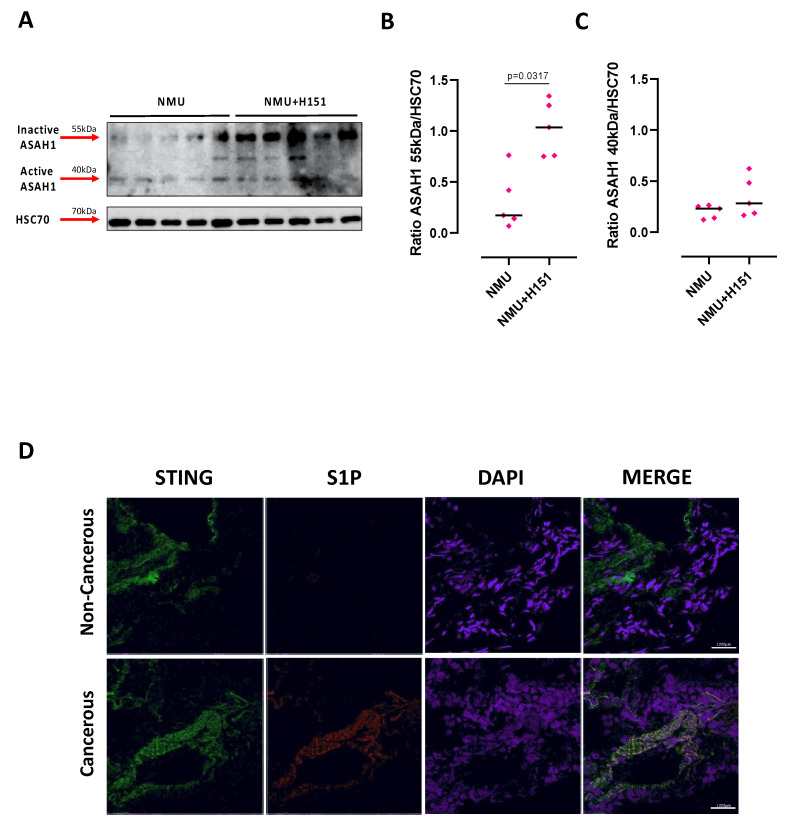
STING and S1P co-expression. (**A**) ASAH1 expression was evaluated in the lungs of NMU- and NMU+H151-treated mice. The active form of ASAH1 (40 kDa) was equally expressed in both groups (**C**), while its inactive form (55kDa) was over-expressed in NMU+H151-treated mice compared to NMU-treated mice (**B**). HSC70 was used as loading control. The quantitative analysis was performed by ImageJ software (1.53a, NIH, Bethesda, MD, USA). Data are represented as scatter dot plots indicating the median (confidence interval = 95%). Statistical differences were assessed by means of two-tailed Mann–Whitney U test. (**D**) Representative immunofluorescence pictures of STING and S1P expression was performed on human non-cancerous and cancerous lung tissues. STING was expressed both in non-cancerous and cancerous tissue (green fluorescence), while S1P was over-expressed in lung cancerous (red fluorescence) compared to non-cancerous tissues. STING and S1P were co-expressed only in cancerous tissue (merge of green and red fluorescence). DAPI was used to stain cell nuclei. Magnification: 40×; scale bar: 1000 µm.

**Figure 8 ijms-24-08303-f008:**
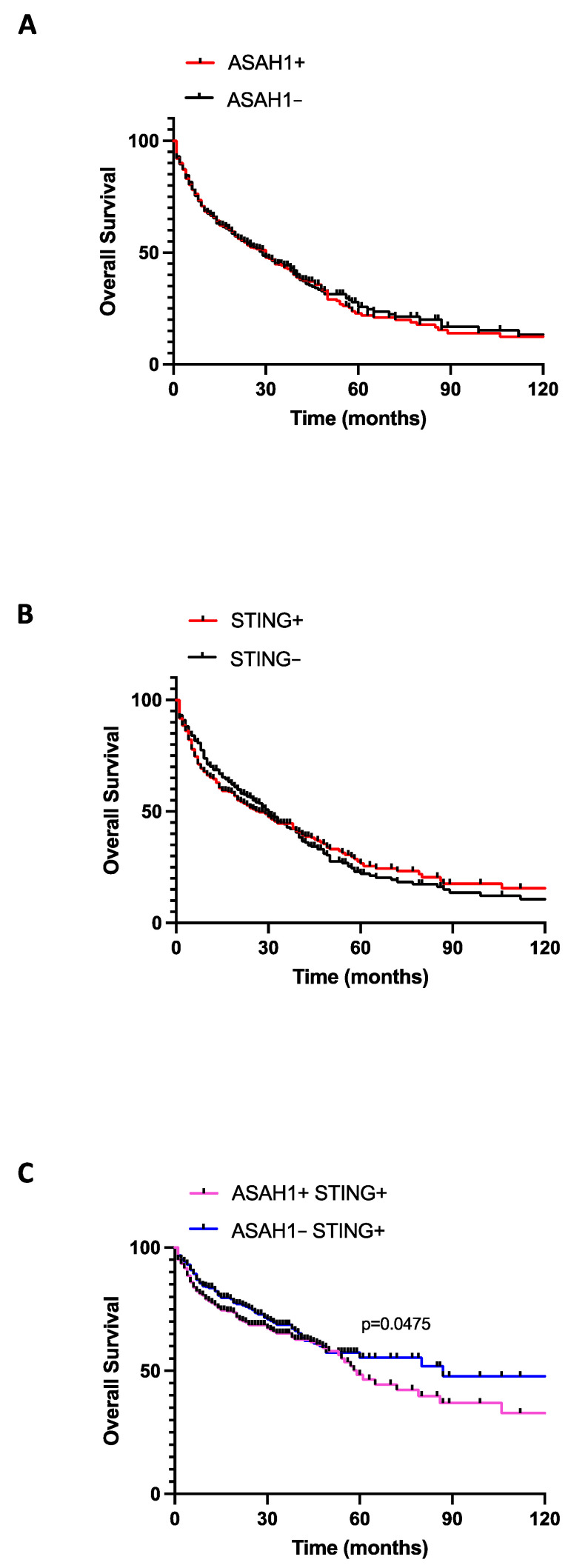
Survival analysis. The public TCGA_LUAD_2016 RNAseq database was used to create a survival curve according to the transcription of ASAH1 (median = 4.643) and STING (median = 2.767) as observed in lung cancer tissues. Patients were stratified in ASAH1+ (*n* = 250) (**A**, red line) and ASAH1− (*n* = 246) (**A**, black line), STING+ (*n* = 248) (**B**, red line) and STING− (*n* = 248) (**B**, black line), and ASAH1+STING+ (*n* = 125) (**C**, pink line) and ASAH1-STING+ (*n* = 98) (**C**, blue line). The overall survival rate was evaluated according to the log-rank test.

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
