# Peer review of "Induction of Inflammation Disrupts the Negative Interplay between STING and S1P Axis That Is Observed during Physiological Conditions in the Lung"

_ijms, 2023, doi:10.3390/ijms24098303_

Round 1

Reviewer 1 Report

Abstract:

STING is not a DNA sensor. It is correctly identified in the Introduction as an adapter protein.

Please correct “reduced to….”.

Results:

Figure 1: Based on the control groups, independent experiments were not performed, so graphs B/D and C/E should be combined for clarity. Figure 1A: Why is the inactive form (55kDa) not detected as it is in Figure 7? Without these data, Figure 1 implies simple upregulation rather than activation.

Lines 115-117: Perhaps this analysis should be moved to the discussion.

Line 129: Confusing sentence: Figures 1 and 2 are performed in lung epithelial cells. This sentence states “Previous data are related to epithelial immortalized tumor cells.”. If this is the correct interpretation, please clarify earlier in the manuscript. The discussion section also states these are tumor cells, although this is not indicated in the results section until line 129.

Figure 2: Based on the control groups, independent experiments were not performed, so graphs A/C and B/D should be combined for clarity.

Please better differentiate the data shown in Figures 3A and 3B and in Figures 3C and 3D. They appear to show the same data and it is important to explain specifically the importance of each image.

Figure 3 legend: Increased mRNA expression could be due to increased stability rather than increased transcription.

Distinguish between “healthy epithelial cells” and “healthy lung tissue”. What cells (other than epithelial cells) are included in the analysis? TCGA_LUAD_2016: What lung tumor types are represented? Can data be analyzed based on lung tumor type? What other tumor-resident cells (other than epithelial cells) are included in the tumor analysis? What was the frequency of ASAH1 +/- and STING +/- in the population? Were the samples chosen for equal numbers of each phenotype, or is approximately 50% +/- normal?

Figure 4: n= 51 or 52. Are these the same samples as Figure 3 (n=58)? Were some samples excluded from these analyses? If so, why?

Line 173-174: “In support, normal fibroblasts in our previous studies did not release inflammatory cytokines compared to cancer cells [14].” This is a very broad statement based on only primary dermal fibroblasts.

Figure 5: Same question as Figure 3. Samples are missing from 5D.

Line 257: Define the acronym NMU. Please include its mechanism of action which may be important to this study.

Lines 261-264: Confusing sentence “We found that in NMU-treated mice the precursor form of the

ceramidase was only slightly induced (ASAH1, 55 kDa, Figure 7A and 7B) while the active form increased compared to the precursor (ASAH1, 40 kDa, Figure 7A; median ASAH1 active form expression=0.23 vs median ASAH1 precursor form expression=0.172), implying its activation.” The western blot and graphs appear to show that STING inhibition highly regulates the _precursor (55 kDa) form. Changes in the active form (40 kDa) are insignificant. See also comment Lines 325-326.

Figures 3B, 4A, 4B, 5B, 6A, 6B: Please label Y axes and heatmaps and GSEA plots.

Discussion:

Lines 325-326: “The lung of mice undergoing NMU treatment had lower activation of ceramidase, thus of S1P, when STING was pharmacologically blocked.” See comment above, Lines 261-264. Please discuss the signaling significance of _inactive_ S1P.

Inhibition of either SPHKI or SPHKII abrogates IL-6 expression. Is phosphorylation of two independent sites necessary for ASAH1 function?

STING protein (maybe RNA) is suppressed in many tumor types.

Figures 3/4, Figures 5/6, although supporting, appear to show simple correlation rather than causation.

Figures 4 and 6 are lacking discussion.

Figure 7 implies that ASAH1 expression relates to carcinogenesis rather than tumor expression. Please discuss this. Also, please discuss why inactive rather than active ASAH1 is so very highly regulated.

Several times the authors state that S1P may control STING and “vice versa”. However, while Figure 1 and 2 show S1P dependence on STING, Figure 7, showing the reverse is less convincing.

With the exception of the Introduction, this submission must be edited for English grammar throughout. The manuscript may be valuable, but parts are difficult to understand.

Reviewer 2 Report

In this review, the authors found that levels of the ceramidase (ASAH1) and STING were inversely correlated in healthy lung epithelial cells, but positively correlated in pathological conditions. The contents are suitable for IJMS. However, there are some major problems to be further improved as well:

1. Authors need to clarify the mechanism of the upregulation expressions of IFN-β or IL-6 after incubating with inhibitors alone including D-NMAPPD, PF543, Opaganib. This is critical for supporting the conclusion of work.

2. The contents and figures need to be carefully improved:

  “precursor enzyme for S1P generation” in line 21 should be moved to the place in line 15.

  “overall survival was reduced to…. Compared to STING/ASAH1 negative patients” in line 22 is difficult to understand.

  Pathway names in Figure 4, 6 need to modified without overlap.

“ASAH1 active form” and “ASAH1 inactive form” need to be added into Figure 7A.

Round 2

Reviewer 1 Report

The authors have addressed this reviewer's comments. Recommend accept.

Reviewer 2 Report

The manuscript is suitable for publication in IJMS.